



# Brief communication: A new methodology for extreme value analysis in the wind industry

Laury Renac[1,4], Marieke Dirksen[1], Bernard Postema[2], Mariska Koning[3], Joseph Jutras[2], and Wouter Pustjens[1]

[1]Pondera Consult a company of Haskoning, 6814 CM Arnhem, The Netherlands
[2]Whiffle, 2629 JD Delft, The Netherlands
[3]Royal Netherlands Meteorological Institute (KNMI), 3731 GA De Bilt, The Netherlands
[4]Aktis Hydraulics, 8011 CS Zwolle, The Netherlands

**Correspondence:** Laury Renac (laury.renac@aktishydraulics.com)

**Abstract.** In the wind industry, estimating extreme wind speeds over a 50-year period using one year of data presents challenges due to the limited representation of extreme events. This study proposes a methodology that combines measurements from Cabauw (The Netherlands) with the ASPIRE 100x100m horizontal resolution large-eddy simulation. Using 20 years of Cabauw measurements as a reference, we compared four methods to estimate the 50-year wind speed ($V_{ref}$). The results show that combining ASPIRE with one year of measurements improves $V_{ref}$ estimates, matching the 20-year reference within 3%. With these promising results, our aim is to apply this methodology to 12-month wind measurement campaigns from the wind industry.

## 1 Introduction

In the wind industry, 12-month wind measurement campaigns are standard for determining long-term wind climates under normal meteorological conditions. However, extreme wind events, critical for wind turbine design, may not occur annually, making it challenging to extrapolate extreme values from one year to a 50-year timescale. While global atmospheric reanalyses like ERA5 or MERRA2 can supplement one year of data, their coarse resolution fails to capture local effects, potentially leading to overdesign or underdesign of wind farms and increasing costs or risks. In this extended abstract, we will describe our novel methodology through which we obtain the 50-year extreme wind speed ($V_{ref}$) using results from large-eddy simulations combined with observations, and compare the results with a.o. wind industry standards.

## 2 Methodology

We propose a novel methodology to obtain the $V_{ref}$ and uncertainty estimates by combining measured storm peaks with high-resolution (100x100m) atmospheric model data from the ASPIRE model (e.g. Baas et al., 2023). ASPIRE resolves local effects, enhancing extreme event representation compared to global models. The study evaluates four methods to estimate $V_{ref}$: (a) 20 years of Cabauw measurements (reference), (b) 1 year of measurements, (c) 20 years of ERA5 data calibrated with 1 year





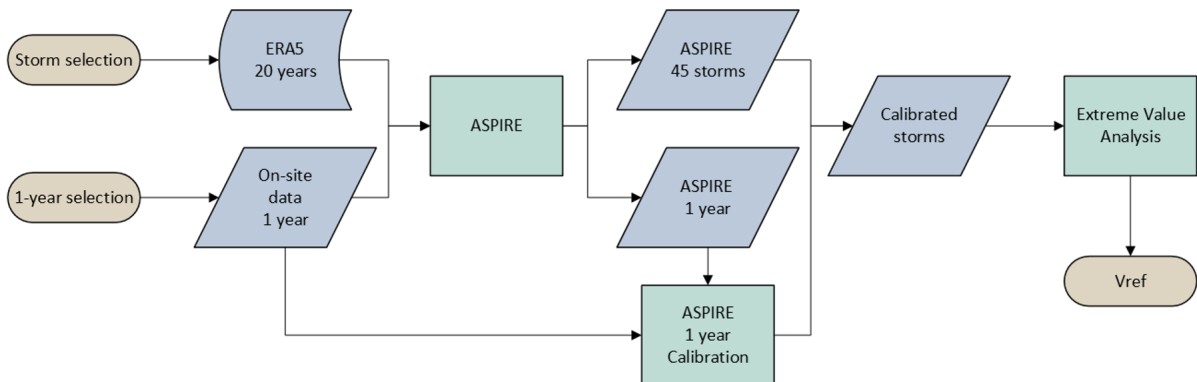

**Figure 1.** Overview of the data processing and analysis proposed novel methodology (d); The storm selection was based on ERA5, from which 45 were simulated with ASPIRE and calibrated with 1 year of measurements.

of measurements, and (d) 45 storms simulated with ASPIRE, calibrated with 1 year of measurements. Figure 1 provides an overview of the data processing and analysis of the proposed novel methodology (d). We discuss the strengths and limitations of each approach.

## 2.1 Wind Measurements

Wind measurements were obtained from the 213 m tall tower at the Cabauw measurement site. The area around the Cabauw site, located near the town Lopik, The Netherlands (51.971 deg N, 4.927 degE), is representative of this part of The Netherlands and is surrounded mostly by pasture and farmland. The terrain is quite homogeneous, especially towards the west, from which the wind prevails the most. At each measurement level, three wind vanes and two wind cups are installed at 120-degree intervals to guarantee sufficient exposure of the sensors at all wind directions.

Long time series of wind measurements along the mast are available, making it suitable as reference measurements for this study. In our analysis, we used the 10-minute averaged wind speed data from 80 m that were obtained between 2001-2020. The data were subjected to automatic validation and visual inspection. Furthermore, flow corrections were applied to correct for mast disturbances (Wessels, 1983). For more information, we refer to the Technical Report of Cabauw Observations (Bosveld, 2024).

### 35 2.1.1 ASPIRE

The model used is the Atmospheric Simulation Platform for Innovation, Research, and Education (ASPIRE). The core of ASPIRE is a large-eddy simulation (LES) code named GRASP (GPU-Resident Atmospheric Simulation Platform). ASPIRE is used for research and commercial purposes by the company Whiffle (The Netherlands). Previous work with ASPIRE includes: Williams et al. (2024), Oldbaum (2019), Baas et al. (2023), Verzijlbergh (2021) (wind farm modelling); Schepers et al. (2021),





**Table 1.** Overview of the model settings for the ASPIRE simulations. The LES domain was nested in the Meso domain.

|  | Coord. | Horiz. points | Horiz. res. | Vert. res. | Top |
|---|---|---|---|---|---|
|  | $^oN, {}^oE$ | $[x_n, y_n]$ | km | m | m |
| Meso | 51.971, 4.927 | 256x256 | 2x2 | 40 | 8000 |
| LES | 51.971, 4.927 | 256x256 | 0.1x0.1 | 25 | 2575 |

Taschner et al. (2023) (turbine physics and loads); Gilbert et al. (2020), Alonzo et al. (2022) (wind forecasting); Kantharaju et al. (2023), Storey and Rauffus (2024) (wind climate modelling); and Bieringer et al. (2021) (dispersion).

In this study, an open-boundary condition LES setup is used, which is nested in a parent mesoscale simulation, which itself takes its boundary conditions from the ERA5 reanalysis (see Table 1 for resolution and domain details). Turbulence in the LES is generated by a series of smaller periodic LESs surrounding the main LES domain (Storey and Rauffus, 2024). The

simulations are done without parametrization for radiation or microphysics. The land surface is parametrized based on the Tile ECMWF Scheme for Surface Exchanges over Land (TESSEL), developed by the ECMWF (ECMWF, 2017). Over water surfaces, the parametrization by Charnock (1955) is applied.

The simulation period encompasses the full year of 2002, and 45 selected 24-hour windows when storms pass. Note that the 24-hour windows are centered around the peak of the storms. Meteorological output variables from the LES are stored as

10-minute averages at the centre of the domain at 80m height.

### 2.1.2  Extreme Value Analysis

We compare the extreme value analysis results from the four methods. The first step of the Peak over Threshold (POT) method was threshold selection; one should make sure that the threshold is high enough to capture rare events but low enough to retain a sufficient sample size. To determine the specific threshold, we used statistical tools such as the Mean Residual Life and an

estimation of the parameter stability based on the threshold. We used an independence criterion of 4 days. Subsequently we fitted a distribution to the extracted peaks, using the Weibull distribution, which is recommended by Det Norske Veritas (DNV) (DNV, 2024) for extreme metocean parameters. Once the distribution was fitted to the extracted peaks, we extrapolated the distribution to the period of interest: the 50 years return period. The POT for the different methods was performed as follows:

(a) 20 years of Cabauw (reference). We applied the POT method, the resulting $V_{ref}$ from this dataset presents the low-
est uncertainties: the measured data is considered highly reliable, and the length of the dataset ensures relatively low extrapolation uncertainties.

(b) 1 year of measurements. We applied the POT to a subset of 1 year of measured data at Cabauw. Considering the very short dataset used, we expect large uncertainties here. We also expect the results to vary significantly depending on the year selected.




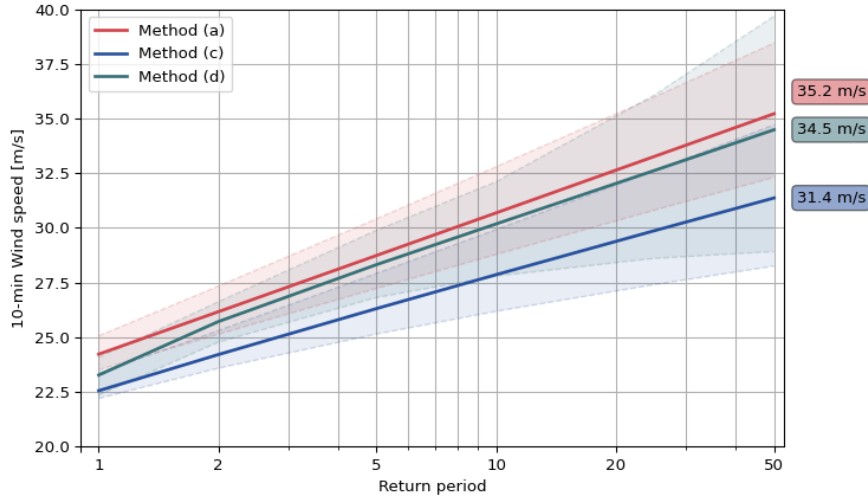

**Figure 2.** Extreme value analysis results for three out of the four methods (a) 20 years of Cabauw (reference), (c) 20 years of ERA5 data calibrated with 1 year of measurements and (d) 45 storms simulated with ASPIRE, calibrated with 1 year of measurements. Method (b) 1 year of measurements was not included due to the large variation in $V_{ref}$ between the years.

(c) 20 years of ERA5 data calibrated with 1 year of measurements. In this case we first used the measured data from 2002 to calibrate the ERA5 data and subsequently performed the POT method on the 20 years of calibrated ERA5 wind speeds. The calibration is done based on an offset and bias correction of the bulk of the data and an additional correction of the highest wind speed values. The ERA5 raw data presents a slight underestimation of the bulk of the data and additional underestimation of the tail. The calibration corrects for both these effects. Subsequently, we apply the POT method to

the calibrated 20 years of ERA5 data.

    (d) 45 storms simulated with ASPIRE, calibrated with 1 year of measurements. For the storm days selection, we used the 45 highest storm peaks from the ERA5 100 m hourly wind speeds with a 24-hour window when the storm passes, excluding the year 2002. The continuous run of the year 2002 was used to obtain calibration constants that we applied to the selected 45 storm days. ASPIRE had a bias (ASPIRE - Cabauw measurements) of -0.1 m/s and a RMSE of 1.7 m/s for

the year 2002. This gave us a dataset of calibrated storm days that we applied the POT method to.

## 3   Results

Figure 2 provides an overview of the $V_{ref}$ from the Extreme Value Analysis (EVA) for three of the four different methodologies:





(a) 20 years of Cabauw (reference). The $V_{ref}$ from the 20 years of measured data at Cabauw is considered to be the reference result. For this case we use a full 20 years of data to extrapolate to a 50-year return period. It gives a $V_{ref}$ of 35.2 m/s and a 90% confidence interval of 32.3-38.5 m/s.

(b) 1 year of measurements (not shown in Figure 2). The $V_{ref}$ based on only one year of measured data at Cabauw is wildly dependent on which year is selected. Depending on the year, $V_{ref}$ results range from 29.9 to 56.9 m/s. For the year 2002, it results in a $V_{ref}$ of 56.5 m/s and a 90% confidence interval of 36.9-153.4 m/s. The higher end of this confidence interval is highly unrealistic and shows the limits of using only a year of data for extrapolation.

(c) 20 years of ERA5 data calibrated with 1 year of measurements. After calibration of the ERA5 data we performed the POT on the 20 years of calibrated data and the resulting fit is shown in Figure 4. It gives a $V_{ref}$ of 31.3 m/s and a 90% confidence interval of 28.2-34.9m/s.

(d) 45 storms simulated with ASPIRE, calibrated with 1 year of measurements. With ASPIRE we performed the POT on the storms and the resulting fit is shown in Figure 2. It gives a $V_{ref}$ of 34.5 m/s and a 90% confidence interval of 28.9-41.9m/s.

## 4  Discussion and Conclusions

We proposed a novel methodology through which we obtained the 50-year extreme wind speed ($V_{ref}$) and uncertainty estimates by combining measured storm peaks with high-resolution data from ASPIRE. ASPIRE had a bias of 0.1 m/s compared to the Cabauw measurements. The 20 years of Cabauw (reference) is considered the most reliable and the length of the dataset ensures relatively low extrapolation uncertainties. We did not account for the climate variability on timescale beyond this measurement period. Using only the year 2002 $V_{ref}$ has a major uncertainty and a non-representative value due to the large number of storms in 2002. The range of $V_{ref}$ for a single year varied between 29.9-56.9 m/s. Compared to using only a single year of measurements, the ERA5 storm peaks improved the extreme value analysis, although compared to the full 20 years from Cabauw the extreme value estimates are 3 m/s lower. 45 storms simulated with ASPIRE, calibrated with 1 year of measurements improved the $V_{ref}$ estimate compared to using only observations. The uncertainty could be further reduced by adding more ASPIRE simulations to the EVA. With our novel methodology the $V_{ref}$ was within 3% of the full 20 years of Cabauw. With these promising results we aim at applying this methodology to 12-month wind measurement campaigns from the wind industry.

*Data availability.*  The Cabauw wind measurements are part of the *cesar_tower_meteo_lc1_t10_v1.0* data set that is freely available from dataplatform.knmi.nl.



*Author contributions.* LR implemented the extreme values analysis, MD and WP conceptualized the content, MK prepared the wind measurements and BP contributed through the ASPIRE simulations. All authors contributed significantly to the writing and reviewing of the paper.

*Competing interests.* No competing interests are present.



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
