# Peer review of "Brief communication: A new methodology for extreme value analysis in the wind industry"

_Wind Energy Science, 2025_

## Referee Comment (RC1)

**General Comments**

The authors address a valid challenge in wind resource and wind project design representing 50-year wind speeds with limited data. They derive return values from four different datasets, two modeled and two measured, with the goal of showing that a comparable $V_{ref}$ can be obtained from a selection of modeled storms as longer-term continuous measurement campaign. The communication is written clearly and succinctly with an emphasis on usability. Some clarifications and refinements to help readers replicate/implement the findings are posed below.

**Specific Comments**

1. WES defines *Brief Communications* for both brevity and impact. Please provide context for designating these findings as unique and high-impact, with justification for its implementation over other methods. 1 year of data in itself does not present a sufficient sample size to calculate a 50-year return value.

2. P. 3: Please specify if the Weibull distribution used is 2-parameter or 3-parameter.

3. Measurements: Please specify the measurement interval (i.e., 10 mins) and any differences (i.e., due to shear) between the 80m measurement values and 100m ERA5 values.

4. P. 3: While DNV guidance may recommend Weibull for "extreme metocean parameters", ocean and atmosphere parameter behavior can differ. This distribution is widely used for wind speeds, however, please provide justification of its suitability at the investigated height for the study location (such as results of a chi squared or Kolmogorov-Smirnov test).

5. P. 4: Please describe the ERA5 highest wind speed calibration choice and method in more detail. What is the threshold to qualify as a high speed?

6. Calibration of two datasets (ERA5 and ASPIRE) appear to be derived from the same 1-year period, and that this year is maintained in the 20-year measurement EVA.

   a. How is interannual variability quantified for the site? How does the calibration treat this—what is the sensitivity to different calibration periods?
   b. Please show that the year used for calibration is representative of long-term conditions at the site. How was the year selected? From the return value estimation on P. 4, it appears that this year was particularly stormy.
   c. How does the $V_{ref}$ for the ASPIRE and Cabauw compare when the training (calibration) year is removed from the measurement dataset (19 instead of 20 years)?

d. In the case of the ASPIRE dataset, it seems that extreme events (45 storms) are calibrated from a continuous sample (i.e., including many normal conditions). Please clarify in the text if this is incorrect, or please justify why these storms are calibrated to the values chosen.

7. Are the 45 storms from the ASPIRE dataset are taken from 1 year? Please show that that measurements and model values, prior to POT analysis, are comparable. Please also describe the range of storms selected (i.e., from X to Y peak winds), and how storms were selected.

8. Please provide more detail on how the ASPIRE dataset is related to the climate, including details on its parent mesoscale simulation and its parameterizations.

9. The argument of the paper would be bolstered by showing similar performance when using different 1-year measurement periods and for different locations (i.e., different from a homogeneous, flat pastured area). If available, please show this.

**Technical Corrections**

1. P. 3: (b) implies that a subset (less than 1 year) of data from the Cabauw campaign is used for calibration. I interpret that 1 year of the Cabauw dataset was selected for calibration. If so, please update for clarity.

---

## Referee Comment (RC3)

**Referee Comments**

Brief communication: A new methodology for extreme value analysis in the wind industry

Laury Renac, Marieke Dirksen, Bernard Postema, Mariska Koning, Joseph Jutras, and

Wouter Pustiens

**A. General Comments:**

**1. Scientific significance:** Does the manuscript represent a substantial contribution to scientific progress within the scope of WES (substantial new concepts, ideas, methods, analyses, or data)?

**R//:**

The problem posed is interesting and relevant for the wind industry, as it addresses the challenge of estimating 50-year extreme wind speeds from short-term measurements. However, the methods presented are not sufficiently rigorous or comprehensive to answer the research question. Although brevity is expected in a Brief Communication, the paper lacks essential methodological details, and the ideas are not well-connected, resulting in limited scientific coherence and flow.

Consequently, I do not consider the manuscript, in its current version, to represent a substantial or novel contribution for readers of *Wind Energy Science*. Please refer to the specific comments below for methodological and conceptual details.

**2. Scientific quality:** Are the scientific approach and applied methods valid? Is sufficient information given so other researchers (in principle) can repeat the work? Are the results discussed in an appropriate and balanced way (consideration of related work, including appropriate references)?

**R//:**

Also related to the previous point, the manuscript presents major methodological concerns, including (i) the preselection of 45 storms, (ii) the questionable application of Extreme Value Theory (EVT) on discrete, non-continuous data, and (iii) the unclear contribution and necessity of scenario (c), which uses 20 years of ERA5 data calibrated with only one year of measurements.

Moreover, the paper does not provide sufficient methodological information for other researchers to replicate the analysis. The discussion of results mainly reports numerical outcomes rather than examining key aspects such as statistical sensitivity, convergence, or physical interpretation.

Given that the topic addressed is of high practical and scientific relevance, it would be more appropriate for this study to be developed in an extended and more complete version as a full Research Article rather than a Brief Communication, in order to cover the missing methodological details and provide a more rigorous scientific foundation.

**3. Presentation quality:** Are the scientific results and conclusions presented in a clear, concise, and well-structured way (abstract conveys efficiently the essence of the paper; number and quality of figures/tables; appropriate, fluent, and precise use of English language)?

**R//:**

The current manuscript lacks depth and clarity in several aspects. Although the writing is understandable, the flow of ideas is irregular, and some concepts are insufficiently defined. For example, the sensitivity of the results to the number of storms is not discussed, and the reasons for selecting specifically 45 storms is not explained. Similarly, the four-day independence criterion is unclear and could be artificially reducing the number of extreme events, thereby increasing the uncertainty of the POT analysis.

In addition, the calibration approach is not neutral or well-justified: if the year 2002 was unusually stormy, calibrating both ASPIRE and ERA5 using that year could amplify the extremes and introduce bias. The a priori selection of 45 storms and their isolated simulation with ASPIRE means that a continuous time series is not used, so the estimated parameters and their confidence intervals may not represent the true asymptotic behaviour of the underlying extreme value distribution. Although preselecting the 45 storms is not necessarily impossible, in that case, the sampling design should be incorporated into the inference, but there is no information about this in the manuscript.

Overall, the manuscript lacks sufficient depth in its presentation of results and in the interpretation of their implications.

**B. Specific Comments:**

- 1) Caption Figure 1: Regarding arbitrariness and the lack of justification of the sample of 45 storms, the selection of the sample size of extremes represents a significant methodological weakness, namely the absence of selection criteria: The authors chose to simulate 45 storms with ASPIRE, based on the 45 highest storm peaks identified in the ERA5 hourly data at 100m, excluding the year 2002. However, the paper does not provide a statistical or practical justification (e.g., computational limitations) for selecting the exact number of 45.
- 2) Line 12 and Caption Fig. 1: It seems there is a contradiction in the preselection of extreme events. The authors point out that reanalyses such as ERA5 have a coarse resolution that "fails to capture local effects," which can lead to overdesign or underdesign of wind farms. By relying on ERA5 to identify storm peaks that will then be simulated at high resolution (ASPIRE), there is a possibility that ERA5 has not correctly identified the most extreme local events, limiting the input dataset for the new methodology (Method d).
- **3)** Lines **65** and **71**: Regarding the selection of the calibration year, calibrating the ERA5 reanalysis and the ASPIRE model using a single year that is recognized as atypical or extreme (2002) could introduce a systematic bias in the correction, potentially affecting the accuracy of 50-year predictions.
- **4) Lines 88-90:** The extrapolation uncertainty is greater than the one from reference, since the 90% confidence interval for V\_ref obtained with Method d was 28.9-41.9 m/s. This range is significantly wider than that obtained with the 20-year reference dataset from Cabauw (32.3-38.5 m/s). This shows that, although the point estimate (V\_ref =34.5 m/s) is good, the 50-year extrapolation from the simulated and calibrated dataset remains more uncertain than the use of long-term measurements. This aspect is not discussed or developed in the paper.
- **5) Line 100:** It might be an insufficient sample of storms. Although the authors acknowledge a limitation in the sample size of ASPIRE simulations, suggesting that "Uncertainty could be further reduced by adding more ASPIRE simulations to the EVA." This indicates that 45 simulated storms may not be statistically sufficient to completely minimize extrapolation uncertainty and limits the validity of extrapolating

extremes to 50 years. Therefore, the paper would benefit from a convergence analysis of the results with 30, 60, or 90 storms in order to detect changes or stabilities with other numbers of storms.

- **6)** Line **27:** Although the ASPIRE model is high resolution (LES, 100x100m) and capable of resolving local effects, there is no discussion on that the calibration constants obtained at a homogeneous site may not be directly transferable to wind farm sites with complex topography or variable coastal/marine effects without a rigorous re-evaluation of the calibration. The study focuses on applying the methodology to 12-month measurement campaigns in the wind industry, but validation is limited to this specific, homogeneous site.
- **7) Line 52:** Note that EVT (and the POT method) not only models the magnitude of extreme events (the shape of the tail), but also the frequency with which peaks above the threshold occur. The way you applied EVT imply a potential methodological risk in occurrence frequency. By using 45 preselected simulations instead of a continuous time series, a distribution of forced events is being modeled. This could compromise the accurate estimation of the annual exceedance frequency (rate parameter), which is crucial for correctly extrapolating to the 50-year return period.
- **8)** Line **55:** Regarding the assumption of independence, and although a 4-day independence criterion is used, this criterion is typically applied to measured data. By simulating only isolated 24-hour windows, the methodology needs to ensure that these 45 storms statistically represent independent peaks, an assumption that can be difficult to guarantee when relying on the selection of a global reanalysis such as ERA5.
- 9) Line 72: The are some implications of vertical extrapolation that accumulate to the other sources of uncertainties. Relying on ERA5 at 100 m to identify the extreme peaks that are then analyzed at 80 m introduces an additional source of uncertainty. This because if the vertical wind shear during extreme storms is strong or non-monotonic, the 45 highest peaks at 100 m may not correspond exactly to the 45 highest peaks at 80 m. Since the subsequent calibration (ASPIRE vs. Cabauw) occurs at 80 m, the reliance on 100 m for the initial filter represents a methodological weakness in the vertical consistency of the extreme event selection process. This and other uncertainty factors are not always recognized, nor are they addressed or discussed rigorously in the paper.
- **10)** Line **73:** There is some doubt as to whether this "global" calibration from 2002 is the most appropriate for correcting the specific behavior of the extreme tail (the 45 storms). The bias and RMSE errors calculated on the complete data set may not be representative of the model's accuracy specifically during the most severe wind events, which are the focus of the analysis.
- **11)** Line **83:** Although 2002 was excluded for calibration purposes, it had a large number of storms. If the proposed methodology (d) is based on combining the physics of the model (ASPIRE) with observations from that calibration year, the exclusion of the most extreme event in the 20-year simulated set could bias the final estimate downward. If the 2002 event is a plausible, albeit rare, extreme, why is it not included in the simulated set to which the EVT is applied?
- **12)** Line **20:** The justification for the need of method c is not provided. It can only be assumed that, although this method is not the main proposal of the study, its inclusion as a comparison scenario helps to justify the need for method d. It would help if its utility is stated in the text.

**C. Technical Comments:**

| 1) Line 86: Please correct the cross reference of the Figure. |
|---------------------------------------------------------------|
|                                                               |
|                                                               |
|                                                               |
|                                                               |
|                                                               |
|                                                               |
|                                                               |
|                                                               |
|                                                               |
|                                                               |
|                                                               |
|                                                               |
|                                                               |
|                                                               |
|                                                               |
|                                                               |